# Effect of Crushing Peanuts on Fatty Acid and Phenolic Bioaccessibility: A Long-Term Study

**DOI:** 10.3390/antiox11020423

**Published:** 2022-02-19

**Authors:** Isabella Parilli-Moser, Inés Domínguez-López, Camila Arancibia-Riveros, María Marhuenda-Muñoz, Anna Vallverdú-Queralt, Sara Hurtado-Barroso, Rosa M. Lamuela-Raventós

**Affiliations:** 1Department of Nutrition, Food Sciences and Gastronomy, School of Pharmacy and Food Sciences XIA, Institute of Nutrition and Food Safety (INSA-UB), University of Barcelona, 08028 Barcelona, Spain; iparillim@ub.edu (I.P.-M.); idominguez@ub.edu (I.D.-L.); carancri77@alumnes.ub.edu (C.A.-R.); mmarhuendam@ub.edu (M.M.-M.); avallverdu@ub.edu (A.V.-Q.); sara.hurtado_17@ub.edu (S.H.-B.); 2CIBER Fisiopatología de la Obesidad y Nutrición (CIBEROBN), Instituto de Salud Carlos III, 28029 Madrid, Spain

**Keywords:** bioavailability, food processing, isoferulic, *p*-coumaric, fatty acids, VLCSFAs

## Abstract

Background: Peanuts are consumed worldwide and have been linked to multiple health benefits. Processing may affect the bioavailability of peanut bioactive compounds. Therefore, we aim to evaluate the effects of crushing peanuts on the bioavailability of fatty acids and phenolic compounds in healthy adults. Methods: 44 participants from the ARISTOTLE study consumed 25 g/day of whole peanuts (WP) or 32 g/day of peanut butter (PB) for 6 months. Fatty acids and phenolic compounds in peanut products and biological samples were assessed by gas chromatography coupled to flame ionization detection and liquid chromatography coupled to high resolution mass spectrometry, respectively. Results: Plasma concentrations of very long chain saturated fatty acids (VLCSFAs) increased significantly after 6 months of WP or PB intake (*p* < 0.001 in both cases). Participants in the WP group excreted twice as many VLCSFAs in feces as those in the PB group (*p* = 0.012). The most abundant polyphenols found in WP and PB were *p*-coumaric and isoferulic acids. Urinary excretion of isoferulic acid increased after the intake of WP and PB (*p* = 0.032 and *p* = 0.048, respectively), with no significant difference observed between interventions. Conclusion: The crushing step in peanut butter production seems to enhance the bioavailability of bioactive compounds.

## 1. Introduction

Peanuts (*Arachis hypogaea*) are botanically classified as legumes, but from a nutritional point of view they are regarded as nuts [1]. In 2018, the global consumption of peanuts increased to 42.6 million metric tons, with over half consumed as peanut butter. The worldwide popularity of these edible seeds is due to their flavor, nutritional contribution, and affordability [2,3].

A higher peanut consumption has been associated with protective effects against cardiovascular diseases (CVD) [4,5,6], including a lower CVD mortality rate [7,8], different types of cancer [9,10], type II diabetes [11,12] and hypertension [6]. In 2003, the Food and Drug Administration stated that following a low-fat diet with 43 g/day of nuts, including peanuts, may reduce the risk of heart disease [13]. These health benefits have been mainly attributed to oleic acid, the predominant monounsaturated fatty acid in peanuts [14,15,16]. In addition, bioactive components such as resveratrol, flavonoids, phenolic acids and phytosterols are reported to have a protective role in health due to their antioxidant and anti-inflammatory properties [14].

The nutritional value of peanuts, as well as the concentration and bioavailability of their bioactive compounds, may be changed by processing [17]. Fat absorption was found to be lower when nuts were consumed whole rather than in the form of butter or oil [18,19].

Studies using in vitro digestion models report that the bioaccessibility of nut lipids is significantly affected by the particle size within the food matrix, and their digestion is limited when cell walls are not disrupted [20,21,22,23]. Additionally, clinical trials have demonstrated that reducing the hardness and particle size of almonds by roasting and chopping enhances lipid digestion and absorption in the small intestine [24,25]. Similarly, cell wall barriers seem to play an important role in regulating polyphenol bioaccessibility [26]. Cell disruption by milling, grinding, and crushing strongly promoted the release of phytochemicals from the food matrix [27,28,29]. Ultra-fine grinding of wheat bran enhanced the bioaccessibility of *p*-coumaric, sinapic, and ferulic acids in bran-enriched bread in an in vitro gastrointestinal model [30]. Moreover, Kuijsten et al. reported a substantial improvement in the bioavailability of lignans in crushed versus whole flaxseeds [31].

However, evidence for the effect of processing on the bioaccessibility and bioavailability of peanut bioactive compounds is still scarce and provided mainly by in vitro or short-term clinical trials. Therefore, to redress this lack of data, the aim of this long-term clinical study was to evaluate the effect of crushing peanuts on the bioavailability of fatty acids and phenolic compounds in healthy young adults.

## 2. Materials and Methods

### 2.1. Study Design

The present study includes data from the ARISTOTLE study, a randomized controlled trial designed to assess the impact of daily peanut and peanut butter intake on the gut microbiota–brain axis, evaluating their pre- and postbiotic effects [32]. Eligible participants were healthy males and females aged 18 to 33 years, with a BMI below 25 kg/m^2^ and without a chronic disease history (cardiovascular diseases, cancer, diabetes mellitus, and others), peanut allergy, or toxic habits such as tobacco smoking or excessive alcohol intake. After approval of the protocol by the Ethics Committee of Clinical Investigation of the University of Barcelona (Barcelona, Spain), the study was registered at https://register.clinicaltrials.gov/ (NCT04324749), accessed on 1 June 2020. Each participant signed an informed consent in advance, according to the principles of the Declaration of Helsinki. Forty-four volunteers who consumed either 25 g/day of whole peanuts (WP) or 2 tablespoons (32 g)/day of peanut butter (PB) for 6 months were included in this study. They were asked to carry out a peanut-free run-in period for two weeks prior to the baseline visit. During the intervention, they followed their habitual diet excluding wine, grapes, dark chocolate (>70%), berries, and nuts. Roasted and unskinned WP were produced in the USA and provided by Ferrer Segarra S.A. (Madrid, Spain), and PB was manufactured from slow-roasted unskinned Virginia peanuts by the Koeze Company (Michigan, MI, USA). Details of the nutritional composition of the test products are described in Table A1 (Appendix A).

### 2.2. Sample Collection

Biological samples were collected at baseline and after 6 months of the intervention. Fasting blood was drawn from the arm via venipuncture into tubes containing EDTA. Serum and plasma were separated after centrifugation at 3000× *g* for 10 min at 4 °C and at 1500× *g* for 15 min at 4 °C, respectively. Participants provided urine from 24 h before each visit. Fecal samples were obtained with the help of a stool collection kit and stored immediately at −20 °C until the visit. All samples were aliquoted and stored at −80 °C until analysis.

### 2.3. Anthropometric, Clinical, and Biochemical Measurements

Anthropometric measurements were taken with the participants in fasting conditions. Height was measured in the standing position using a portable stadiometer. Weight and body composition (body fat and muscle percentages) were measured using a tetrapolar OMRON BF511 bioelectrical device, with the participants wearing light clothes and no shoes. BMI was calculated as weight divided by height squared (kg/m^2^). Waist and hip circumferences were measured using an inelastic flexible tape positioned equidistantly between the lowest rib and the iliac crest and on upper trochanters, respectively. The waist-to-hip ratio was calculated by dividing the waist by the hip circumference. Blood pressure was measured in a sitting position in triplicate using an OMRON M6 digital monitor. Biochemical markers in serum and plasma (glucose and lipid profile, respectively) were measured in an external laboratory (Cerba Internacional, Barcelona, Spain) using enzymatic methods.

### 2.4. Dietary Intake and Physical Activity

Diet and physical activity were recorded by trained staff at baseline and at the end of the intervention using a validated 151-item semi-quantitative food frequency questionnaire (FFQ) and a Spanish validated version of the Minnesota Leisure-Time Physical Activity Questionnaire, respectively [33,34].

### 2.5. Determination of Fatty Acids in Peanut Products, Plasma and Feces

#### 2.5.1. Reagents and Standards

Sodium methylate, boron trifluoride in methanol (14% *w*/*v*) and *n-hexane* were purchased from Sigma-Aldrich (St. Louis, MO, USA), sodium chloride from Panreac Quimica SLU (Barcelona, Spain) and anhydrous sodium sulphate from Schalab (Barcelona, Spain). Tridecanoic acid methyl ester (C13:0), used as an internal standard, was purchased from Sigma-Aldrich, and the standards Supelco 37 Component FAMEs mix and PUFAs No. 2 (Animal source) from Merck (Darmstadt, Germany). Standards were stored in powder form at –20 °C and protected from the light.

#### 2.5.2. Sample Preparation

Before the analysis, derivatization of compounds to their corresponding FAMEs was carried out based on previously described methods for plasma [35], feces [36] and WP and PB [37]. First, the internal standard, tridecanoic acid methyl ester (C13:0; 20 µL for plasma and 40 µL for feces, WP and PB) was added to 200 µL of plasma, 100 mg of fecal sample, and 50 mg of WP and PB. A total of 1 mL of sodium methylate (0.5% *w*/*v*) was added and the mixture was heated to 100 °C for 15 min. After cooling, the samples were esterified at 100 °C for 15 min using 1 mL of boron trifluoride-methanol reagent (14% *v*/*v*). Subsequently, FAMEs were isolated by adding 500 μL (for plasma) and 1 mL (for feces, WP and PB) of *n-hexane*. After shaking for 1 min, 1 mL of a saturated sodium chloride solution was added to the biological samples and 2 mL to the food products. Finally, the tubes were centrifuged for 10 min at 3000 rpm. After drying with anhydrous sodium sulphate, the clear *n-hexane* top layer was transferred into an automatic injector vial equipped with a volume adapter of 300 μL.

#### 2.5.3. Chromatographic Analysis

Fast-gas chromatographic analysis was performed using a Shimadzu GC-2010 Gas Chromatograph (Shimadzu, Kyoto, Japan) equipped with a flame ionization detector and a Shimadzu AOC-20i Autoinjector. Separation of FAMEs was carried out on a capillary column (40 m × 0.18 mm i.d. × 0.1 µm film thickness) coated with an RTX-2330 stationary phase of 10% cyanopropyl phenyl-90% byscyanopropyl polysiloxane from Restek (Bellefonte, PA, USA). 

Operating conditions were as follows: the split-splitless injector was used in split mode with a split ratio of 1:50, the injection volume of the sample was 1 µL, and the injector and detector temperatures were kept at 250 °C and 300 °C, respectively. The temperature program was as follows: initial temperature 110 °C, increased at 52 °C/min to 195 °C, held at this temperature for 6 min, then increased at 25 °C/min to 230 °C, and held for 6.5 min. Hydrogen was used as the carrier gas at a constant pressure of 26 psi, with a linear velocity of 40 cm/s at 110 °C.

Data acquisition and processing were performed with the Shimadzu-Chemstation software for gas chromatography systems. Methyl ester peaks were identified by comparison of their retention times with the standards Supelco 37 Component FAMEs mix and PUFAs No. 2. Results were expressed as plasma, fecal, and food fatty acid concentrations (in µg/mL, µg/100 mg, and mg/g, respectively). The fatty acids quantified were the following: myristic acid (C14:0), pentadecanoic acid (C15:0), palmitic acid (C16:0), palmitoleic acid (C16:1 n-7), heptadecanoic acid (C17:0), stearic acid (C18:0), vaccenic acid (C18:1 n7), oleic acid (C18:1 n-9), linoleic acid (C18:2 n-6), alpha-linolenic acid (C18:3 n-3), gamma-linolenic acid (C18:3 n-6), arachidic acid (C20:0), eicosenoic acid (C20:1 n-9), eicosadienoic acid (C20:2 n-6), eicosapentanoic acid (C20:5 n-3), arachidonic acid (C20:4 n-6), behenic acid (C22:0), erucic acid (C22:1 n-9), docosatetraenoic acid (C22: 4 n6), docosapentaenoic acid (C22:5 n-3), docosahexaenoic acid (C22:6 n-3), tricosanoic acid (C23:0), lignoceric acid (C24:0), and tetracosenoic acid (C24:1 n-9).

### 2.6. Determination of Phenolic Compounds in Peanut Products and Urine

#### 2.6.1. Extraction and Quantification of Phenolic Compounds from Peanut Products

##### Reagents and Standards

Gallic, caffeic, protocatechuic, sinapic, 2,5-dihydroxybenzoic, 2,6-dihydroxybenzoic, *m*-coumaric, *o*-coumaric, *p*-coumaric acids and resveratrol and rutin were purchased from Sigma-Aldrich (St. Louis, MO, USA); ferulic, isoferulic, and vanillic acid from Fluka (Buchs, Switzerland); and kaempferol-3-O-glucoside and quercetin-3-O-glucuronide from Extrasynthese (Genay, France). Formic acid, ACN, and ethanol (EtOH) were acquired from Merck (Darmstadt, Germany). Ultrapure water was obtained from a Milli-Q water purification system (Millipore Bedford, MA, USA). All samples (WP and PB) and standards were treated in a room with light filters to avoid polyphenol oxidation.

##### Sample Preparation 

The extraction of polyphenols was performed following a previously reported procedure with minor modifications [38]. A total of 0.5 g of WP and PB were homogenized with a blender and then mixed with 5 mL of 80% EtOH in Milli-Q water (0.1% formic acid) and vortexed for 1 min. The mixture was then sonicated in an ultrasound bath with ice to prevent overheating for 20 min and centrifuged at 4000 rpm for 20 min at 4 °C. The supernatant was collected in a tube and the extraction procedure was repeated. The supernatants were combined and evaporated under a nitrogen flow and the residue was reconstituted with Mili-Q water (0.1% formic acid) up to 4 mL. The extract was filtered by a 0.45 µm polytetrafluoroethylene filter into an insert amber vial. Samples were stored at −20 °C until analysis by liquid chromatography coupled to linear trap quadrupole-Orbitrap mass spectrometry (LC–LTQ-Orbitrap-MS).

##### Chromatographic Analysis

Liquid chromatography analysis was carried out in an Accela chromatograph (Thermo Scientific, Hemel Hempstead, UK) equipped with a quaternary pump and a thermostated autosampler. Chromatographic separation was performed using a Kinetex C18 column (2.1 × 150 mm, 3.5 µm) acquired from Waters (Milford, MA, USA). Gradient elution was performed with water/0.1% formic acid (solvent A) and ACN/0.1% formic acid (solvent B) at a constant flow rate of 0.350 mL/min, and the injection volume was 10 µL. A non-linear gradient was used: 0 min, 5% B; 1 min, 5% B; 7 min, 45% B; 8.5 min, 80% B; 10.5 min, 80% B; 11 min, 5% B; 12 min, 5% B, and the column was equilibrated for 5 min prior to each analysis. The LC system was coupled to an LTQ-Orbitrap Velos mass spectrometer (Thermo Scientific, Hemel Hempstead, UK), used for accurate mass measurements and equipped with an ESI source operated in negative mode. Operation parameters were as follows: source voltage, 4 kV; sheath gas, 20 a.u. (arbitrary units); auxiliary gas, 10 a.u; sweep gas, 2 a.u; and capillary temperature, 275 °C. Samples were analyzed in Fourier transformation mass spectrometry (FTMS) mode at a resolving power of 30,000. Parent ions were fragmented by high-energy C-trap dissociation (HCD) with a normalized collision energy of 35 V and an activation time of 10 min. The mass range in both modes was from *m*/*z* 100 to 2000. Instrument control and data acquisition were performed with Xcalibur 3.0 software (Thermo Fisher Scientifi, Hemel Hempstead, UK). Quantification of polyphenols was carried out using the internal standard method. Polyphenols were quantified with respect to their corresponding standard.

#### 2.6.2. Extraction and Quantification of Phenolic Acids in Urine

##### Reagents and Standards

The pool of standards was prepared in synthetic urine: protocatechuic acid, *m*-coumaric acid, *o*-coumaric acid, *p*-coumaric acid, and dihydroresveratrol from Sigma-Aldrich (St. Louis, MO, USA); and vanillic acid from Fluka (Buchs, Switzerland). All reagents were of HPLC grade: EtOH, ACN, methanol (MeOH) and formic acid were purchased from Merck (Darmstadt, Germany). Ultrapure water (Milli-Q) was obtained from a Millipore system (Millipore, Bedford, MA, USA).

##### Sample Preparation

Phenolic compounds were extracted from the urine samples by solid phase extraction (SPE) using a previously published procedure with minor modifications [39]. Prior to the SPE, urine samples were diluted 1:20, acidified with formic acid, and centrifuged at 14,000 rpm for 4 min at 4 °C. Then, MeOH (1 mL) and 1.5 M formic acid (1 mL) were added to activate the HLB plate 30 µm (30 mg). A total of 950 µL of urine, previously acidified as explained and spiked with abscisic acid (IS), were loaded into the 96-well plate for clean-up with 1.5 M formic acid (1 mL) and 0.5% MeOH solution (1 mL). The elution was achieved with MeOH (1 mL) acidified with 0.1% formic acid. The elution fraction obtained was evaporated to dryness by a sample concentrator (Techne, Staffordshire, UK) at room temperature under a stream of nitrogen. A total of 100 µL of water acidified with 0.05% formic acid was added to dissolve the residue and filtered through a 0.22 µm polytetrafluoroethylene syringe filter (Waters Corporation, Milford, MA, USA).

##### Chromatographic Analysis

Chromatographic analysis of phenolic compounds was performed according to a previously validated method [39] adapted to an LTQ-Orbitrap Velos mass spectrometer (Thermo Scientific, Hemel Hempstead, UK) equipped with an ESI source operated in negative mode. The separation was carried out with Milli-Q water and can, with 0.05% formic acid in both solvents, using a Kinetex 2.6 µm Column (50 mm × 4.6 mm) acquired from Phenomenex (Torrence, CA, USA). Quantification of polyphenols was carried out using the internal standard method. Polyphenols were quantified with respect to their corresponding standards.

### 2.7. Statistical Analyses

Analysis of variance was used to compare the means of fatty acids and phenolic compounds from WP and PB products. For biological samples, normality of distribution was assessed by a Shapiro–Wilk test. Because of the small sample size (<30 in each group) and non-normal distribution, non-parametric tests were used. A Wilcoxon matched-pair signed-rank test was used to evaluate differences between baseline and 6 months within each group. Changes in plasma/fecal fatty acids and urinary phenolic compounds (post-intervention value minus the baseline value) were calculated, and the resulting differences were tested by the Wilcoxon rank-sum test. Categorical variables are expressed as number (n) and proportion (%) and continuous variables as mean ± SD. Statistical analyses were assessed for individual fatty acids and for their subtypes according to the saturation degree and series as: SFAs (C14:0, C15:0, C16:0, C18:0, C:20, C:22, C23:0 and C:24), MUFAs (C16:1 n-9, C16:1 n7, C18:1 n-7, C18:1 n-9, C20:1 n-9, C22:1 n-9 and C24:1 n-9), PUFAs (C18:2 n-6, C18:3 n-6, C18:3 n-3, C18:3 n-6, C20:2 n-6, C20:5 n-3, C20:4 n-6, C22:4 n-6, C22:5 n-3 and C22:6 n-3), and VLCSFAs (C20:0, C22:0, and C24:0). Differences were considered significant when the *p* value was lower than 0.05. Statistical analyses were conducted using STATA software version 15.

## 3. Results and Discussion

### 3.1. Participant Characteristics

Table 1 shows the baseline characteristics of the 44 healthy participants included in this sub-study. Subjects had an average age of 22.28 ± 3.13 years, their average BMI was 22.15 ± 3.06 kg/m^2^, and 50% had finished a 4-year college or graduate course. The mean of reported physical activity was higher than 4000 METs/week. No significant differences between groups were found in any of the variables.

### 3.2. Dietary Intake and Physical Activity

After 6 months of intervention, a significant reduction in physical activity was reported by participants consuming WP compared to baseline (*p* = 0.012), but no significant differences were observed between groups (Table A1). Since the study was focused on the bioavailability of polyphenols and fatty acids from peanuts, as these dietary components are widespread in food [40] it was important to assess the overall food and nutrient intake of the participants and how it might influence the results. No significant differences were found in nutrient intake among the young adults taking part. As expected, the consumption of nuts was significantly lower after both WP and PB interventions compared to baseline, given that the participants were asked to exclude nuts from their diet due to their having a similar lipid content to peanuts [41]. In addition, a trend towards a higher olive oil consumption was observed in WP versus PB consumers (*p* = 0.064).

### 3.3. Fatty Acid Profile in Whole Peanuts and Peanut Butter

The fatty acid content of WP and PB, expressed as a relative percentage and total amount per mg of sample, is presented in Table 2. The main fatty acid in both foods was oleic acid, which represented 55–61% of the total fat content (339 and 364 mg/mg of WP and PB, respectively), followed by linoleic acid (24% and 20% for WP and PB, respectively). Similar findings were reported by Hinds et al., Orsavova et al., and Maguire et al.—although the latter study found higher concentrations of linoleic acid than oleic acid [42,43,44]. Meanwhile, the main saturated fatty acids in both WP and PB were palmitic and stearic acids, in accordance with the data of Maguire et al., who reported the fatty acid profile of five different types of nuts, including peanuts [43]. Very long chain saturated fatty acids (VLCSFAs), including arachidic, behenic, and lignoceric acid, accounted for 6 and 5% of the total fat in WP and PB, respectively. Peanuts are reported to be a good source of VLCSFAs [45,46], which are therefore good candidate biomarkers of peanut consumption—unlike oleic, linoleic, palmitic, and stearic acids, all of which are widely distributed in foods. Although the fatty acid composition can be altered by crushing, with variable effects according to the technological process used, we found that WP and PB had similar profiles. Significant differences were only found in levels of palmitoleic, eicosadienoic and tricosanoic acids, which were higher in WP compared to PB (*p* = 0.003, *p* < 0.001 and *p* = 0.002, respectively).

### 3.4. Fatty Acid Bioavailability in Participants

#### 3.4.1. Plasma Fatty Acids

As shown in Table 3, significantly higher plasma concentrations of total VLCSFAs (*p* < 0.001 in both groups), arachidic (*p* = 0.020 and *p* = 0.012, respectively), behenic (*p* < 0.001 in both groups), and lignoceric acids (*p* < 0.001 in both groups) were observed after the WP and PB intervention compared to baseline, without differences between the two groups, which correlates with the similar fatty acid profile found in the tested products (Table 2). These results are consistent with those from a similar nutritional trial where peanut butter intake led to higher plasma concentrations of behenic, lignoceric, and cerotic acid after 2–8 h of consumption [46]. The beneficial properties of these VLCSFAs have not been extensively investigated. To date, research on the effects of arachidic, behenic and lignoceric acid on type 2 diabetes has not yielded consistent results [45,47,48]. Otherwise, an inverse association between these fatty acids in plasma and erythrocytes and cardiovascular health outcomes has been reported [49,50,51]. In the present study, the plasma concentrations of the two major fatty acids—oleic and linolenic acid—increased significantly after WP consumption, but decreased after PB intake (*p* = 0.040 and *p* = 0.049, respectively). However, as these fatty acids are present in a wide variety of foods, this increase cannot be attributed exclusively to the peanut intervention. For example, the increase in olive oil consumption by participants in the WP group, a product rich in oleic acid, could have affected the results (Table A1). Regarding other fatty acids, gamma-linolenic acid increased, whereas arachidonic acid decreased in plasma after the WP intervention compared to baseline (*p* = 0.038 and *p* = 0.049, respectively)—but without significant differences between the two groups. In the case of palmitic and stearic acids, despite their high content in peanut products, no increase in plasma levels was observed.

#### 3.4.2. Fecal Fatty Acids

The fatty acid profile in the feces of the participants is provided in Table 4. As in plasma, fecal concentrations of total VLCSFAs increased significantly after the WP and PB interventions compared to baseline (*p* < 0.001 and *p* = 0.009 of WP and PB, respectively). Notably, participants in the WP group excreted twice as many VLCSFAs, as well behenic acid and lignoceric acid, compared to the PB group (*p* = 0.012, *p* = 0.006 and *p* = 0.026 and, respectively).

Regarding the other saturated fatty acids, WP consumers excreted significantly more stearic acid than PB consumers (*p* = 0.026), as well as more total saturated fatty acids (*p* = 0.040); the former also excreted significantly more palmitic acid post-intervention compared to baseline (*p* = 0.007), although without a significant difference from the PB group. The results suggest that the main saturated fatty acids in whole peanuts are not bioavailable and were eliminated through the feces. Higher concentrations of monounsaturated fatty acids (*p* = 0.043 and *p* = 0.003 for WP and PB, respectively) and oleic acid (*p* = 0.036 and *p* = 0.007 for WP and PB respectively) were also observed in feces after the two interventions compared to baseline. Fecal levels of oleic acid, which did not differ significantly between groups, reflected the content of oleic acid determined in the peanut products and the plasma samples from the WP group.

In agreement with our results, other intervention studies comparing the effects of WP and PB intake report higher levels of fecal fat in WP consumers [18,52], as do studies on different forms of almonds [21,53,54]. Fat bioaccessibility can be affected by interactions with other food components, such as dietary fiber [55]. Moreover, the low bioavailability of peanut fatty acids is attributed to the resistance of cell walls, which act as a physical barrier against the action of lipase and reduce the bioaccessibility of lipids and energy extraction [56].

By affecting particle size, the processing of nuts alters lipid bioaccessibility [19,57,58]. Thus, the crushing involved in peanut butter production may be expected to improve fatty acid bioavailability, even though this was not observed in the plasma fatty acid results. A reason for the lack of statistical differences between the two groups could be the very low plasma concentrations of arachidic, behenic, and lignoceric acid.

### 3.5. Polyphenol Composition of Whole Peanuts and Peanut Butter

As shown in Table 5, phenolic acids, mainly *p*-coumaric and isoferulic acids, were the most abundant polyphenols in WP and PB, representing more than 60–70% of the total polyphenols, with a higher concentration in PB than WP (*p* = 0.037 and *p* = 0.008, respectively). Previous studies have similarly reported that *p*-coumaric and its derivatives are the major phenolic compounds in raw peanuts, with variable contents according to the cultivar [59,60]. 

Many factors could explain the differences in the phenolic content of peanut products, including environmental growing conditions, germination, ripening, harvest time, processing, storage conditions and the extraction method [61,62,63]. Peanuts are a well-known source of antioxidants and polyphenols, whose concentration is highest in their skins [62,64]. It has been observed that PB produced with added skins, such as the one used in the present study, has a higher antioxidant capacity and phenolic content than PB made without skin [65,66]. Nevertheless, both products that we administered contained skins, with crushing being the main distinguishing factor. Similar to our results, Sobolev et al. found a significantly higher resveratrol content in peanut butter than in roasted peanuts. In addition, Yu et al. determined several classes of phenolics in peanut skins, such as phenolic acids, stilbenes, and flavonoids—the latter being the most abundant, in contrast with our results [67].

### 3.6. Polyphenol Bioavailability in Participants 

#### Urinary Polyphenols

Vanillic acid was the predominant phenolic compound found in urine after both interventions (Table 6). Regarding the two main polyphenols quantified in the peanut products, isoferulic acid and *p*-coumaric acid, only the former increased significantly in the urine of WP and PB consumers after the intervention compared to baseline (*p* = 0.030 and *p* = 0.048 respectively). However, these phenolic compounds are widely distributed in foods other than peanuts. The absence of significant differences between WP and PB interventions could be due to high standard deviations, which can hinder statistical treatment, as has been observed in previous studies [68,69,70].

Our results could have been influenced by inter-individual variability, including sex/gender differences, health status and gut microbiota composition, which may affect the bioavailability, metabolism, and excretion of polyphenols [71]. Studies with adults and adolescents have demonstrated that males excrete more polyphenols than females [72,73]. In animal models, the greater polyphenol excretion found in males was associated with a higher expression of UDP-glucuronosyltransferase, one of the enzymes responsible for polyphenol absorption in the small intestine [74]. However, in the present study, no significant gender-dependent differences were observed (data not shown), possibly due to the high proportion of female participants (73%). Another important factor of this variability is the composition and activity of the gut microbiota, which is involved in the metabolism and absorption of polyphenols that cannot be absorbed in the small intestine [75]. According to some studies, polyphenol bioaccessibility may be reduced or delayed by binding with dietary fibers [76,77], and therefore phenolic compounds in fiber-rich foods more readily reach the lower parts of the digestive tract. Polyphenols are metabolized into more bioavailable compounds by intestinal enzymes, and in turn can modulate the gut microbial balance, with beneficial health effects [78].

It has also been hypothesized that crushing processes during food production promote the liberation of phytochemicals during digestion due to the disruption and breakdown of cellular components [79]. If the cell wall in peanuts is not ruptured, their antioxidants are lost in the feces [58]. Overall, it is difficult to generalize about the bioavailability of a specific phytochemical, as each case is different—detailed analysis of the food matrix is required [26]. In addition, as urinary excretion data can underestimate polyphenol bioavailability [17], it is of interest to determine levels in feces as well as microbial metabolites in order to achieve a more complete and accurate picture.

## 4. Conclusions

This study provides evidence for the lower digestibility of lipids from whole peanuts compared to peanut butter, as fecal levels of very long chain saturated fatty acids, mainly found in peanut products, were significantly higher after the consumption of whole peanuts. This result suggests that the crushing step in peanut butter processing may enhance the bioavailability of these fatty acids, which have been associated with protective effects against chronic diseases. Furthermore, after the consumption of whole peanuts, saturated fatty acids also seemed to be lost in feces rather than absorbed. Another benefit of the crushing process was an enhanced phenolic content in the food product, although no differences in phenolic bioavailability were observed. Further research into the bioavailability of bioactive food components and the underlying mechanisms will enable the design of new functional foods and provide support for health claims.

## Figures and Tables

**Table 1 antioxidants-11-00423-t001:** Participant characteristics at baseline.

	Whole Peanuts (n = 21)	Peanut Butter (n = 23)	*p*-Value
**Female, n (%)**	14 (66)	18 (78)	0.388
**Education level, n (%)**			0.652
University students	11 (52%)	11 (48%)	
Graduated	10 (48%)	12 (52%)	
**Age (years)**	22.28 ± 3.20	23.43 ± 2.90	0.142
**Body composition**			
Weight (kg)	63.26 ± 10.12	60.10 ± 7.72	0.240
BMI (kg/m^2^)	22.12 ± 3.52	22.19 ± 2.59	0.541
Waist circumference (cm)	72.73 ± 8.31	71.28 ± 5.53	0.796
Hip circumference (cm)	98.74 ± 6.35	95.85 ± 6.24	0.120
Waist to hip ratio	0.73 ± 0.06	0.74 ± 0.05	0.415
Body fat (%)	26.66 ± 8.07	28.45 ± 7.88	0.404
Muscle mass (%)	32.09 ± 5.71	31.04 ± 5.81	0.459
**Physical activity (METs/week** **)**	4850 ± 2124	4703 ± 2382	0.751
**Blood pressure**			
SBP (mmHg)	112 ± 7.34	110 ± 8.87	0.235
DBP (mmHg)	72.63 ± 7.63	72.87 ± 6.20	0.698
**Blood analytes**			
Glucose (mmol/L)	4.54 ± 0.44	4.59 ± 0.35	0.760
Triglycerides (mmol/L)	0.71 ± 0.20	0.85 ± 0.35	0.152
Total cholesterol (mmol/L)	4.33 ± 0.52	4.60 ± 0.88	0.318
LDL-cholesterol (mmol/L)	2.22 ± 0.39	2.60 ± 0.69	0.070
HDL-cholesterol (mmol/L)	1.75 ± 0.30	1.69 ± 0.40	0.459
**Dietary intake**			
Energy (kcal/day)	2771 ± 594	2706 ± 602	0.816
Carbohydrates (g/day)	257 ± 80.74	241 ± 73.92	0.642
Protein (g/day)	104 ± 29.43	110 ± 31.86	0.388
Total fat (g/day)	145 ± 29.17	142 ± 35.35	0.816
SFAs (g/day)	37.62 ± 10.00	38.18 ± 11.05	0.514
MUFAs (g/day)	67.76 ± 15.90	69.06 ± 17.18	0.852
PUFAs (g/day)	25.91 ± 6.77	23.99 ± 7.25	0.499

Data are expressed as mean ± standard deviation (SD). BMI: body mass index; METs: metabolic equivalents of task; SBP: systolic blood pressure; DBP: diastolic blood pressure; LDL-cholesterol: low density lipoprotein cholesterol; HDL-cholesterol: high density lipoprotein cholesterol; SFAs: saturated fatty acids; MUFAs: monounsaturated fatty acids; PUFAs: polyunsaturated fatty acids. The *p*-value column refers to differences between groups at baseline. *p* values < 0.05 are statistically significant and were calculated by the Wilcoxon rank-sum test for continuous variables and the Chi-square test for categorical variables.

**Table 2 antioxidants-11-00423-t002:** Fatty acid profile in whole peanuts and peanut butter.

Fatty Acids	Whole Peanuts		Peanut Butter		*p*-Value
	%	mg FA/g Peanut	%	mg FA/g Peanut Butter	
Myristic acid (C14:0)	0.03	0.16 ± 0.02	0.02	0.14 ± 0.01	0.101
Palmitic acid (C16:0)	9.84	60.59 ± 8.07	8.38	49.71 ± 3.61	0.100
Palmitoleic acid (C16:1 n-7)	0.08	0.48 ± 0.05	0.05	0.27 ± 0.01	0.003
Heptadecanoic acid (C17:0)	0.10	0.59 ± 0.08	0.09	0.55 ± 0.02	0.447
Stearic acid (C18:0)	3.02	18.61 ± 2.45	2.81	16.69 ± 1.17	0.288
Oleic acid (C18:1 n-9)	55.11	339 ± 45.90	61.49	364 ± 27.48	0.461
Linoleic acid (C18:2 n-6)	24.23	149 ± 20.36	20.24	120 ± 9.22	0.086
Arachidic acid (C20:0)	1.44	8.90 ± 1.18	1.38	8.19 ± 0.57	0.406
Alpha-linolenic acid (C18:3 n-3)	0.05	0.31 ± 0.04	0.04	0.25 ± 0.02	0.081
Eicosenoic acid (C20:1 n-9)	1.11	6.86 ± 0.93	1.19	7.05 ± 0.49	0.768
Eicosadienoic acid (C20:2 n-6)	0.10	0.61 ± 0.08	0.02	0.10 ± 0.02	<0.001
Behenic acid (C22:0)	3.11	19.15 ± 2.58	2.64	15.66 ± 1.03	0.095
Erucic acid (C22:1 n-9)	0.08	0.52 ± 0.09	0.08	0.47 ± 0.03	0.423
Tricosanoic acid (C23:0)	0.10	0.63 ± 0.10	0.04	0.24 ± 0.02	0.002
Lignoceric acid (C24:0)	1.52	9.40 ± 1.34	1.43	8.50 ± 0.48	0.335
**SFAs**	19.16	118 ± 15.81	16.81	99.67 ± 6.90	0.139
VLCSFAs	6.18	38.07 ± 5.19	5.50	32.59 ± 2.10	0.165
**MUFAs**	56.42	347 ± 46.99	62.85	372 ± 28.02	0471
**PUFAs**	24.42	150 ± 20.51	20.34	120 ± 9.26	0.083
**Total fatty acids**		616 ± 83.31		593 ± 44.18	0.634

Data are expressed as mean ± standard deviation (SD). SFAs: saturated fatty acids; VLCSFAs: very long chain saturated fatty acids; MUFAs: monounsaturated fatty acids; PUFAs: polyunsaturated fatty acids. Statistical analyses were carried out using Student’s t-tests. *p*-values refer to differences between whole peanut and peanut butter concentration. *p*-values < 0.05 were considered significant and are shown in bold.

**Table 3 antioxidants-11-00423-t003:** Plasma fatty acids of healthy adults consuming whole peanuts or peanut butter in the ARISTOTLE study.

Plasma Fatty Acids (µg/mL)	Whole Peanuts			Peanut Butter			*p*-Value WP vs. PB
	Baseline	Post- Intervention	Post-Intervention–Baseline	Baseline	Post- Intervention	Post-Intervention–Baseline	
Myristic acid (C14:0)	15.46 ± 5.89	16.05 ± 7.74	0.59 ± 6.17	21.62 ± 16.81	19.72 ± 11.17	−2.76 ± 16.74	0.355
Palmitic acid (C16:0)	519 ± 275	541 ± 172	22.50 ± 285	643 ± 194	651 ± 165	8.12 ± 224	0.128
Palmitoleic acid (C16:1 n-7)	41.80 ± 14.78	47.26 ± 15.59	5.47 ± 14.99	55.01 ± 33.51	87.61 ± 132.38	28.80 ± 134.57	0.878
Stearic acid (C18:0)	190 ± 64.88	216 ± 150	25.65 ± 134	221 ± 46.80	212 ± 59.05	−18.01 ± 60.73	0.200
Vaccenic acid (C18:1 n-7)	32.24 ± 12.04	40.50 ± 10.08	8.26 ± 12.61	49.31 ± 16.46	78.38 ± 124	25.66 ± 123	0.474
Oleic acid (C18:1 n-9)	469 ± 169	544 ± 143	74.45 ± 135	691 ± 155	669 ± 207	−51.38 ± 203	**0.040**
Linoleic acid (C18:2 n-6)	757 ± 272	886 ± 215	129 ± 186	999 ± 194	982 ± 280	−60.04 ± 325	**0.049**
Alpha-linolenic acid (C18:3 n-3)	22.35 ± 53.82	8.53 ± 4.23	−13.82 ± 54.10	13.48 ± 9.92	11.56 ± 4.25	−2.43 ± 8.75	0.630
Gamma-linolenic acid (C18:3 n-6)	6.47 ± 4.48	8.41 ± 4.24 *	1.94 ± 3.64	10.12 ± 4.48	53.23 ± 204	40.79 ± 201	0.065
Arachidic acid (C20:0)	2.54 ± 1.32	3.06 ± 1.03 *	0.51 ± 1.51	3.06 ± 1.01	4.23 ± 1.58 *	1.12 ± 1.66	0.264
Eicosenoic acid (C20:1 n-9)	7.39 ± 7.66	6.63 ± 3.03	−0.76 ± 8.65	9.65 ± 7.77	40.59 ± 88.16	29.17 ± 86.44	0.378
Eicosadienoic acid (C20:2 n-6)	8.30 ± 8.90	7.75 ± 2.70	−0.55 ± 8.84	9.25 ± 3.29	8.52 ± 3.57	−1.11 ± 4.27	0.148
Eicosapentanoic acid (C20:5 n-3)	25.60 ± 20.54	24.69 ± 9.91	−0.91 ± 19.11	28.04 ± 19.65	44.93 ± 50.75	14.94 ± 53.92	0.235
Arachidonic acid (C20:4 n-6)	321 ± 159	202 ± 51.45 *	−119 ± 120	222 ± 59.51	214 ± 75.14	−17.21 ± 80.51	0.062
Behenic acid (C22:0)	2.09 ± 1.03	4.66 ± 1.69 *	2.57 ± 1.71	2.63 ± 1.51	5.33 ± 1.52 *	2.58 ± 2.26	0.953
Docosatetraenoic acid (C22: 4 n-6)	17.78 ± 16.92	16.21 ± 5.42	−1.57 ± 18.51	18.76 ± 3.19	17.99 ± 3.22	−1.54 ± 5.67	0.062
Docosapentaenoic acid (C22:5 n-3)	21.97 ± 13.64	20.15 ± 4.45	−1.82 ± 15.63	22.68 ± 5.08	22.21 ± 5.14	−1.44 ± 7.35	0.431
Docosahexaenoic acid (C22:6 n-3)	69.42 ± 25.55	81.06 ± 22.47	11.65 ± 30.77	85.65 ± 33.33	92.80 ± 38.31	3.12 ± 44.40	0.474
Lignoceric acid (C24:0)	3.45 ± 0.77	9.28 ± 3.93 *	5.83 ± 3.81	3.93 ± 2.16	11.18 ± 4.02 *	6.93 ± 4.18	0.318
Tetracosenoic acid (C24:1 n-9)	11.44 ± 6.06	11.31 ± 3.68	−0.13 ± 8.00	13.34 ± 3.01	13.09 ± 5.74	−0.82 ±7.51	0.431
**SFAs**	755 ± 215	769 ± 351	12.65 ± 255	854 ± 254	895 ± 214	39.50 ± 204	0.990
VLCSFAs	8.09 ± 2.54	17.00 ± 5.70 *	8.92 ± 5.37	9.62 ± 3.48	20.73 ± 5.55 *	11.11 ± 5.69	0.157
**MUFAs**	563 ± 185	650 ± 167	87.29 ± 152	819 ± 199	889 ± 306	66.26 ± 293	0.507
**PUFAs**	1240 ± 847	1247 ± 273	7.37 ± 844	1402 ± 232	1439 ± 253	37.11 ± 243	0.222
**Total fatty acids**	2558 ± 949	2666 ± 663	107 ± 615	3116 ± 593	3043 ± 907	66.88 ± 539	0.419

Data are expressed as mean ± standard deviation (SD). WP: whole peanuts; PB: peanut butter; SFAs: saturated fatty acids; VLCSFAs: very long chain saturated fatty acids; MUFAs: monounsaturated fatty acids; PUFAs: polyunsaturated fatty acids. *p*-values WP vs. PB refer to differences between whole peanut and peanut butter consumers regarding changes in fatty acid levels (post-intervention value minus the baseline value) calculated by Wilcoxon rank-sum tests. *p*-values < 0.05 were considered significant and are shown in bold. *: refers to significant differences between baseline and 6 months (post-intervention) results within each group, assessed by the Wilcoxon signed-rank test.

**Table 4 antioxidants-11-00423-t004:** Fecal fatty acids of healthy adults consuming whole peanuts or peanut butter in the ARISTOTLE study.

Fecal Fatty Acids (µg/100 mg)	Whole Peanuts			Peanut Butter			*p*-Value WP vs. PB
	Baseline	Post- Intervention	Post-Intervention–Baseline	Baseline	Post- Intervention	Post-Intervention–Baseline	
Myristic acid (C14:0)	38.68 ± 41.81	59.51 ± 60.79	20.83 ± 65.41	30.99 ± 18.28	95.57 ± 256	64.57 ± 255	0.842
Pentadecanoic acid (C15:0)	59.65 ± 49.12	83.15 ± 85.52	23.49 ± 93.10	54.42 ± 32.78	49.61 ± 32.82	−4.81 ± 41.10	0.534
Palmitic acid (C16:0)	730 ± 791	1108 ± 766 *	378 ± 724	578 ± 277	936 ± 828	358 ± 825	0.445
Palmitoleic acid (C16:1 n7)	33.03 ± 65.92	45.63 ± 90.08	12.60 ± 34.91	14.01 ± 6.35	25.52 ± 26.52	11.51 ± 25.56	0.935
Heptadecanoic acid (C17:0)	31.60 ± 30.29	45.82 ± 64.29	14.22 ± 67.00	21.36 ± 9.77	24.66 ± 13.33	3.30 ± 14.73	0.823
Stearic acid (C18:0)	1074 ± 130	1885 ± 408	811 ± 605	1232 ± 784	1132 ± 1110	−100 (733)	**0.026**
Oleic acid (C18:1 n9)	593 ± 664	2202 ± 551 *	1680 ± 308	376 ± 523	1250 ± 994 *	873 (137)	0.264
Vaccenic acid (C18:1 n7)	188 ± 740	90.47 ± 218	−97.62 ± 526	31.09 ± 22.91	45.00 ± 62.29	13.90 (66.19)	0.663
Linoleic acid (C18:2 n6)	348 ± 592	660 ± 977	312 ± 792	349 ± 697	587 ± 702	238 (883)	0.860
Arachidic acid (C20:0)	31.45 ± 26.45	61.57 ± 39.17 *	30.11 ± 32.52	25.30 ± 10.74	41.13 ± 22.67 *	15.84 (22.27)	0.103
Alpha-linolenic acid (C18:3 n3)	48.16 ± 92.41	55.26 ± 116	7.10 ± 133	160 ± 397	42.13 ± 68.34	−118 (383)	0.445
Eicosenoic acid (C20:1 n9)	16.11 ± 17.47	35.97 ± 7.92 *	19.86 ± 42.22	7.51 ± 4.60	19.53 ± 17.45 *	12.02 (18.12)	0.664
Eicosadienoic acid (C20:2 n6)	5.87 ± 5.52	7.31 ± 10.54	1.43 ± 9.93	5.12 ± 9.30	4.06 ± 5.72	−1.06 ± 10.70	0.630
Behenic acid (C22:0)	26.72 ± 26.95	76.13 ± 39.37 *	49.41 ± 41.60	21.35 ± 11.53	42.18 ± 32.67 *	20.83 ± 33.55	**0.006**
Arachidonic acid (C20:4 n6)	3.54 ± 2.53	4.01 ± 3.47	0.47 ± 3.35	0.87 ± 0.72	1.59 ± 1.90	0.72 ± 2.23	0.953
Eicosapentanoic acid (C20:5 n3)	2.48 ± 4.80	1.14 ± 1.21	−1.33 ± 4.99	1.03 ± 1.58	4.70 ± 11.33	3.67 ± 11.47	0.053
Docosatetraenoic acid (C22: 4 n6)	10.55 ± 9.77	13.84 ± 11.70	3.29 ± 9.19	20.53 ± 9.46	20.62 ± 10.29	0.09 ± 10.47	0.378
Lignoceric acid (C24:0)	24.81 ± 24.50	54.21 ± 28.51 *	29.39 ± 34.47	21.39 ± 8.33	35.49 ± 18.85 *	14.10 ± 20.96	**0.026**
Tetracosenoic acid (C24:1 n9)	6.92 ± 3.81	10.77 ± 11.51	3.84 ± 12.02	7.93 ± 4.23	7.58 ± 4.57	−0.68 ± 6.91	0.341
**SFAs**	2017 ± 1775	3975 ± 4997 *	1957 ± 1609	1986 ± 1036	2357 ± 1924	371 ± 1623	**0.040**
VLCSFAs	82.99 ± 73.05	192 ± 102 *	108 ± 106	68.03 ± 28.51	119 ± 69.06 *	50.76 ± 71.99	**0.012**
**MUFAs**	838 ± 1199	2385 ± 5872 *	1547 ± 4873	437 ± 538	1347 ± 2075 *	910 ± 2217	0.727
**PUFAs**	419 ± 684	742 ± 1043	324 ± 850	538 ± 821	661 ± 805	123 ± 1183	0.889
**Total fatty acids**	3274 ± 2953	7102 ± 1439 *	3828 ± 2777	2960 ± 1832	4366 ± 3216	1405 ± 7362	0.560

Data are expressed as mean ± standard deviation (SD). WP: Whole peanuts; PB: peanut butter; SFAs: saturated fatty acids; VLCSFAs: very long chain saturated fatty acids; MUFAs: monounsaturated fatty acids; PUFAs: polyunsaturated fatty acids. *p*-values WP vs. PB refer to differences between whole peanut and peanut butter consumers regarding changes in fatty acid levels (post-intervention value minus the baseline value) calculated by Wilcoxon rank-sum tests. *p*-values < 0.05 were considered significant and are shown in bold. *****: refers to significant differences between baseline and 6 months (post-intervention) results within each group, assessed by the Wilcoxon signed-rank test.

**Table 5 antioxidants-11-00423-t005:** Polyphenol content of whole peanuts and peanut butter.

Polyphenols (mg/100 g)	Whole Peanuts	Peanut Butter	*p*-Value WP vs. PB
**Flavonoids**	0.99 ± 0.04	9.38 ± 0.64	**0.002**
Catechin	0.23 ± 0.01	3.24 ± 0.13	**<0.001**
Epicatechin	0.21 ± 0.02	5.65 ± 0.71	**0.005**
Quercetin 3-β-d-glucoside	0.02 ± 0.00	0.01 ± 0.00	**0.001**
Quercetin 3-*O*-glucuronide	0.03 ± 0.00	0	**0.001**
Kaempferol-*O*-glucoside	0.08 ± 0.00	0.07 ± 0.01	0.237
Rutin	0.41 ± 0.01	0.40 ± 0.03	0.719
8-prenylnaringenin	0.01 ± 0.00	0.01 ± 0.01	0.290
**Stilbenes**			
Resveratrol	0.32 ± 0.01	0.34 ± 0.02	0.432
**Phenolic acids**	47.82 ± 1.34	98.69 ± 11.26	**0.015**
Protocatechuic acid	1.52 ± 0.09	0.26 ± 0.03	**<0.001**
2,5-dihydroxybenzoic acid	0.16 ± 0.14	0.38 ± 0.05	0.098
Vanillic acid	1.90 ± 0.09	0.88 ± 0.10	**<0.001**
Caffeic acid	0.49 ± 0.05	0.26 ± 0.02	**0.011**
*p*-coumaric acid	24.53 ± 0.35	41.15 ± 3.33	**0.037**
*m*-coumaric acid	1.65 ± 0.03	1.59 ± 0.05	0.357
*o*-coumaric acid	0.06 ± 0.00	0.02 ± 0.00	**0.002**
Ferulic acid	2.13± 0.08	1.71 ± 0.18	**0.040**
Isoferulic acid	13.91 ± 0.60	48.81 ± 5.53	**0.008**
Sinapic acid	1.35 ± 0.09	1.80 ± 0.31	0.116

Data are expressed as mean ± standard deviation (SD). WP: Whole peanuts; PB: peanut butter. Statistical analyses were carried out using Student’s t-tests. *p*-values refer to differences between whole peanuts and peanut butter. *p*-values < 0.05 were considered significant and are shown in bold.

**Table 6 antioxidants-11-00423-t006:** Urinary polyphenols of healthy adults consuming whole peanuts or peanut butter in the ARISTOTLE study.

Urine Polyphenols (mg/day)	Whole Peanuts			Peanut Butter			*p*-Value WP vs. PB
	Baseline	Post- Intervention	Post-Intervention–Baseline	Baseline	Post- Intervention	Post-Intervention–Baseline	
**Stilbenes**							
Dihidroresveratrol	0.05 ± 0.18	0.01 ± 0.01	−0.04 ± 0.18	0.02 ± 0.01	0.03 ± 0.02	0.01 ± 0.02	0.397
**Phenolic Acids**							
Protocatechuic acid	1.13 ± 0.86	1.10 ± 1.14	−0.02 ± 1.17	2.08 ± 1.80	1.72 ± 1.71	−0.35 ± 2.21	0.787
2,5 dihidroxibenzoic	1.09 ± 0.78	1.18 ± 1.17	0.08 ± 1.08	1.87 ± 1.79	1.71 ± 1.75	−0.16 ± 2.11	0.935
Vanillic acid	8.10 ± 6.02	6.63 ± 5.78	−1.46 ± 4.89	16.69 ± 25.82	13.69 ± 21.19	−3.33 ± 14.70	0.860
*p*-Coumaric acid	0.54 ± 1.34	0.37 ± 0.33	−0.18 ± 1.30	0.18 ± 0.21	0.43 ± 0.47	0.23 ± 0.50	0.431
*m*-Coumaric acid	0.53 ± 0.95	0.36 ± 0.57	−0.17 ± 0.76	0.39 ± 1.12	0.40 ± 0.49	0.01 ± 1.23	0.378
*o*-Coumaric acid	0.33 ± 0.84	0.12 ± 0.14	−0.21 ± 0.82	0.19 ± 0.22	0.22 ± 0.30	0.03 ± 0.18	0.431
Isoferulic	0.78 ± 0.64	1.49 ± 1.22 *	0.71 ± 1.11	0.79 ± 0.53	2.95 ± 4.32 *	2.17 ± 4.26	0.916
Sinapic acid	0.58 ± 0.68	0.38 ± 0.47	−0.20 ± 0.77	4.23 ± 12.84	8.36 ± 27.73	4.14 ± 14.94	0.200

Data are expressed as mean ± standard deviation (SD). WP: whole peanuts; PB: peanut butter. *p*-values WP vs. PB refer to differences between whole peanut and peanut butter consumers regarding changes in polyphenol levels (post-intervention value minus the baseline value) calculated by Wilcoxon rank-sum tests. *p*-values < 0.05 were considered significant. *: refers to significant differences between baseline and 6 months (post-intervention) results within each group, assessed by the Wilcoxon signed-rank test.

## Data Availability

The data presented in this study are available in this manuscript.

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
