# Peer review of "Effect of Crushing Peanuts on Fatty Acid and Phenolic Bioaccessibility: A Long-Term Study"

_antioxidants, 2022, doi:10.3390/antiox11020423_

Round 1

Reviewer 1 Report

The authors carried out a 6-month dietary intervention study to evaluate the impact of the consumption of two products, peanuts and peanut butter on the levels of fatty acids in the blood and faeces of the volunteers and on the levels of certain components of the phenolic fraction in the urine of the participants in the study. The underlying hypothesis is that due to the different matrix characteristics of the selected products, resulting from the processing of the products, the bioaccessibility of the selected compounds will be higher in the peanut butter and, consequently their bioavailability would be higher in this product.

The main conclusion of the study is that the results do not clearly support this hypothesis. 

In my opinion, to understand the variability of the observed results it is necessary to consider the composition of the diets that the volunteers followed during the study. The authors indicate in section 2.1 that the volunteers followed their usual diet during the study, and data of these diets are gathered in Table 1. As can be seen, the daily amount of ingested fat is 140-145 g. The intervention introduces 25 g of peanuts or 32 g of peanut butter in the diet of the volunteers. This amount of product represents, appropriately, between 12.5 and 16 g of extra fat in the diet, this value falls within the variability of fat intake in the diets of the volunteers. That it is a rather small quantity and daily fat intake variability would hamper the interpretation of the results. For example, for oleic and linoleic fatty acids (the main fatty acids of peanuts) their levels in blood show statistical differences between the two groups of volunteers. However, the differences favour the group consuming whole peanuts instead of those eating peanut butter.

In my opinion, a wider discussion of this point is needed throughout the manuscript and should also be included in the conclusion section of the text.

Additional comments:

Following section 3.1 of the manuscript, section 3.2 is strange since the description refers to a table (A1) that is not present in the manuscript and includes conclusions on the results obtained after the six months of intervention. However, section 3.1 discusses the characteristics of the individuals before starting the study.

Regarding the composition of the two products used in the study, a table with the data of the proximals is missing. Section 3.3 shows data on the fatty acid composition of the products, but this information should be accompanied by the data of the proximals. Also, the units in which the fatty acid composition data are shown are a bit odd since they are given in mg fatty acid / mg of product, it is difficult to understand how there can be 500-600 mg of fatty acids in one mg of product.

As for section 3.4.1 there is again a reference to table A1 which is not in the pdf of the article.

Author Response

Dear Editors,

We sincerely thank the editor and all reviewers for the opportunity to have our manuscript reviewed in Antioxidants, and for their valuable suggestions. We have tried to address all the concerns and questions from reviewers.  

We provide bellow a point-by-point reply to the reviewers’ comments, and explanation of the changes included in the manuscript (in blue). As it is requested, the revised versions have been submitted as a document with track changes.

We hope that the reviewed manuscript will be considered to be published in Antioxidants. 

Kind regards, 

Rosa M Lamuela-Raventós

Reviewer 1: 

In my opinion, to understand the variability of the observed results it is necessary to consider the composition of the diets that the volunteers followed during the study. The authors indicate in section 2.1 that the volunteers followed their usual diet during the study, and data of these diets are gathered in Table 1. As can be seen, the daily amount of ingested fat is 140-145 g. The intervention introduces 25 g of peanuts or 32 g of peanut butter in the diet of the volunteers. This amount of product represents, appropriately, between 12.5 and 16 g of extra fat in the diet, this value falls within the variability of fat intake in the diets of the volunteers. That it is a rather small quantity and daily fat intake variability would hamper the interpretation of the results. For example, for oleic and linoleic fatty acids (the main fatty acids of peanuts) their levels in blood show statistical differences between the two groups of volunteers. However, the differences favour the group consuming whole peanuts instead of those eating peanut butter. In my opinion, a wider discussion of this point is needed throughout the manuscript and should also be included in the conclusion section of the text.

We would like to thank Reviewer 1 for the global appreciation of our manuscript and suggestions provided.

As an answer to the question, the amount of peanut and peanut butter was chosen according to the portions from United States Department of Agriculture (USDA) and World Health Organization (WHO). USDA defines a portion of peanuts and peanut butter as 1 oz (28 g) and 2 Tbsp (32 g) respectively. In Spain, a handful of nuts (around 25 g) is widely considered as a recommended portion.

As we mentioned in the manuscript our participants follow their usual diet. In table 1 we showed the main macronutrients intake at baseline to assure that there are not difference between groups before starting the intervention. After that we showed at table A1 (in the revised version of manuscript change to Table A2) the changes of the diet (nutrient and food intake) after six-month consuming peanut and peanut butter. Since it is important to assess the overall food and nutrient intake of the participants and how it might influence our results. We are aware that in the peanut group, there is an increase in both Oleic and Linoleic acid compared to peanut butter group. However, these two fatty acids, despite being in high proportion in our products, are also found in many other foods. Therefore, we have analyzed the consumption of specific foods and nutrients that may be affecting our results and we have found that the participants from the peanut group tend to consume more olive oil, the main source of oleic acid in the diet, (p-value: 0.064 – Table A2) this may be promoting the higher levels of oleic acid in those participants consuming peanuts (discussed in lines 304 – 310).

Nevertheless, one strength of our study is that despite fatty acids are presented in different foods, we have focused and found interesting results in three fatty acids exclusive of peanut intake (C20:0, C22:0 and C24:0)

Additional comments:

Following section 3.1 of the manuscript, section 3.2 is strange since the description refers to a table (A1) that is not present in the manuscript and includes conclusions on the results obtained after the six months of intervention. However, section 3.1 discusses the characteristics of the individuals before starting the study.

As the reviewer 1 mentioned, in the section 3.1 we introduced the baseline characteristics of our participants, and the following section 3.2 present the results of the dietary intake and physical activity at baseline and after six months. Since the study is focused on the bioavailability of bioactive compounds widely spread in the diet, we consider important to evaluate the food and nutrient intake during the intervention and these results are discussed in section 3.2 (lines 219-269) and shown in Table A1 (in the revised version of the manuscript change to Table A2) placed at the end of the manuscript in the appendix section before the references` section following the guidelines of the journal (line 447).

Regarding the composition of the two products used in the study, a table with the data of the proximals is missing. Section 3.3 shows data on the fatty acid composition of the products, but this information should be accompanied by the data of the proximals.

Thanks to reviewer 1 suggestion, a table with data of the proximals available from the two products was added to the manuscript, it is place at the appendix section as Table A1 (line 446). 

Also, the units in which the fatty acid composition data are shown are a bit odd since they are given in mg fatty acid / mg of product, it is difficult to understand how there can be 500-600 mg of fatty acids in one mg of product. As for section 3.4.1 there is again a reference to table A1 which is not in the pdf of the article.

As the reviewer mentioned the unit in which the fatty acid composition of food products is shown in table 2 is wrong. As it is indicated in line 151, the results of fatty acid concentration in the food products are expressed as mg/g of product. Therefore, we have corrected the unit expression at table 2. Moreover, we corroborated that the rest of units were correctly mentioned.

Reviewer 2 Report

This is a very interesting manuscript which describes the fate and impact of long term peanuts consumption in the form of whole peanut and peanut butter in the human body. The study is well designed according to the ethical clearance with clear indication of informed consents etc. Scientifically it is very interesting domain which describes the long effect of a very common  nutraceutical i.e. peanuts. the experiments are well conducted and results are well described and supported by the relevant literature. It is however not clear if the study involved only female participants (as shown in table 1) or also the male participants (section 3.6.1, line 381). If the study was conducted only on female volunteers please explain why the male volunteers were excluded from the study. if the male volunteers were not excluded, then why is there no data shown about them? Table 1, please separate the different sections since in the percentage section, the numbers do not add up to 100% but rather more than 100%. please reformat in a way that reader does not get confused with the numbers and split into different zones. 

minor comments. please use italics for the name of plant and in the chemical formulae (for example n-hexane). Page numbers got lost after page 9, please correct it.   

Author Response

Dear Editors,

We sincerely thank the editor and all reviewers for the opportunity to have our manuscript reviewed in Antioxidants, and for their valuable suggestions. We have tried to address all the concerns and questions from reviewers.  

We provide below a point-by-point reply to the reviewers’ comments, and explanation of the changes included in the manuscript (in blue). As it is requested, the revised versions have been submitted as a document with track changes.

We hope that the reviewed manuscript will be considered to be published in Antioxidants. 

Kind regards, 

Rosa M Lamuela-Raventós

Reviewer #2

This is a very interesting manuscript which describes the fate and impact of long term peanuts consumption in the form of whole peanut and peanut butter in the human body. The study is well designed according to the ethical clearance with clear indication of informed consents etc. Scientifically it is very interesting domain which describes the long effect of a very common  nutraceutical i.e. peanuts. The experiments are well conducted and results are well described and supported by the relevant literature. It is however not clear if the study involved only female participants (as shown in table 1) or also the male participants (section 3.6.1, line 381). If the study was conducted only on female volunteers please explain why the male volunteers were excluded from the study. If the male volunteers were not excluded, then why is there no data shown about them?

We would like to thank Reviewer #2 for the global appreciation of our manuscript and suggestions provided.

Regarding the volunteers included, our study involved female and male volunteers. To avoid readers' misunderstandings, we have changed the line 71 in methods section to: “Eligible participants were healthy males and females aged 18 to 33 years…”.

The analyses of the results shown in the manuscript treated men and women data together without stratifying by gender, since there were no significant differences between males and female (mentioned in line 380).

Table 1, please separate the different sections since in the percentage section, the numbers do not add up to 100% but rather more than 100%. please reformat in a way that reader does not get confused with the numbers and split into different zones.

In the table 1 there are two categorical variables (gender and education level) which are expressed as number (n) and proportion (%), we have reordered the table in order that the first variables are categorical and the rest numerical, in that way we consider that it will be easier to follow for the readers.

Regarding the sum of percentages, the proportions belong to each group per separate. In the case of the education level, 52% of the participants in the whole peanut group are university students and the remaining 48% are graduated, which sum 100%.

In the case of the gender, we show only female. Therefore, the proportion of females in the whole peanut group is 66% (14 of the 21 participants) being the remaining percentage (34%) the proportion of males in that group. While in the peanut butter group, 78% (18 of the 23 participants) are females and the remaining (22%) are males.

Minor comment:

Please use italics for the name of plant and in the chemical formulae (for example n-hexane).

As suggested, the name of plants and chemical formulae throughout the manuscript were changed to italic format.

Page numbers got lost after page 9, please correct it.  

The page numbers have been corrected.

Round 2

Reviewer 1 Report

Please check lines 261 and 308, use Table A2 instead of Table A1